# Profiling DNA break sites and transcriptional changes in response to contextual fear learning

**Ryan T. Stott**[1,2], **Oleg Kritsky**[1,2], **Li-Huei Tsai**[1,2]*

**1** Picower Institute for Learning and Memory, Massachusetts Institute of Technology, Cambridge, MA, United States of America, **2** Department of Brain and Cognitive Sciences, Massachusetts Institute of Technology, Cambridge, MA, United States of America

* lhtsai@mit.edu

**Data Availability Statement:** All sequencing data used in this manuscript can be found in the Gene Expression Omnibus repository; sequencing generated for this manuscript can be found at accession number GSE155095 and previously

## Abstract

Neuronal activity generates DNA double-strand breaks (DSBs) at specific loci *in vitro* and this facilitates the rapid transcriptional induction of early response genes (ERGs). Physiological neuronal activity, including exposure of mice to learning behaviors, also cause the formation of DSBs, yet the distribution of these breaks and their relation to brain function remains unclear. Here, following contextual fear conditioning (CFC) in mice, we profiled the locations of DSBs genome-wide in the medial prefrontal cortex and hippocampus using γH2AX ChIP-Seq. Remarkably, we found that DSB formation is widespread in the brain compared to cultured primary neurons and they are predominately involved in synaptic processes. We observed increased DNA breaks at genes induced by CFC in neuronal and non-neuronal nuclei. Activity-regulated and proteostasis-related transcription factors appear to govern some of these gene expression changes across cell types. Finally, we find that glia but not neurons have a robust transcriptional response to glucocorticoids, and many of these genes are sites of DSBs. Our results indicate that learning behaviors cause widespread DSB formation in the brain that are associated with experience-driven transcriptional changes across both neuronal and glial cells.

## Introduction

Neuronal activity has been reported to generate DSBs [1–6]. This was initially observed in cultured neurons, where a well-known marker of DSBs, γH2AX (phosphorylation on serine 139 of histone H2A variant X [7]), rapidly increased following glutamate receptor activation [2]. Subsequently, stimulation of the rodent brain was found to generate DSBs following seizures [5] or behavioral manipulation [1, 3]. While wakefulness in zebrafish [4], or wakefulness with exploration in fruit flies and mice [8], increased DSBs in neurons that were reduced during sleep.

One source of genomic stress in the brain is its high transcriptional output; neurons respond in real-time to environmental changes and this activity necessitates continual modulation of transcription [9]. We made the unexpected discovery that stimulating the activity of

published sequencing can be found at accession number GSE74971.

**Funding:** This work was supported by research grants from National Institutes of Health https://www.nih.gov/ (R01NS102730-01), the Glenn Foundation for Medical Research https://glennfoundation.org/ and the JPB Foundation https://www.jpbfoundation.org/ to LHT. RS was the recipient of the MIT Presidential Fellowship, the Barbara Weedon Fellowship, and the Lord Foundation Fellowship. The funders had no role in study design, data collection and analysis, decision to publish, or preparation of the manuscript.

**Competing interests:** The authors have declared that no competing interests exist.

primary cortical neurons generates DSBs specifically at the rapidly induced early response genes (ERGs), and this promotes their expression [3]. Increases in γH2AX at some of these ERGs was later observed in the brain during fear learning [6] or following memory retrieval [10]. In other contexts of gene induction, including through transcriptional induction mediated by nuclear receptors [11–14] or heat shock and serum-stimulation [15], DSBs appear to facilitate gene induction. Within the complex milieu of the brain, it is therefore likely that different upstream pathways contribute to the generation of DSBs, yet their locations and their relation to brain function is an open question. As DSBs pose a threat to genomic integrity [3], understanding the genome-wide DSB landscape of the brain would facilitate our understanding of how the brain balances timely transcriptional responses with the generation of DSBs, while revealing sites of genomic stress that could seed DNA lesions detrimental to neuronal function and contribute to brain aging and neurodegenerative diseases.

We set out to understand the *in vivo* landscape of DSBs in the brain during learning and how they correspond with gene expression changes occurring in neurons and glia. We find fear learning paradigm-induced genes are overrepresented amongst those genes with the highest levels of DSBs in the medial prefrontal cortex and hippocampus. These genes are downstream of pathways that are shared in part by neurons and non-neurons, and in other cases unique to each group of cells. Surprisingly, we find potential glia-enriched DSB hotspots at genes that have a robust transcriptional response to glucocorticoid receptor signaling in glia.

## Results

### Fear learning induces DNA double-strand breaks in the brain

Increases in neuronal activity result in the formation of DSBs both *in vitro* and *in vivo* [1, 3]. However, it was unclear whether DSBs form at specific genomic loci in the brain and in which cell types in response to a normal physiological event. To elicit neuronal activation in a physiologically relevant manner, we utilized contextual fear conditioning (CFC), which generates a strong associative memory between a novel environment and an aversive stimulus, a foot shock [16]. We assessed neuronal activation in the hippocampus (HIP) and the medial prefrontal cortex (mPFC) of adult wild-type male C57BL/6J mice, two brain regions known to be recruited during CFC for subsequent memory formation [16]. Induced expression of ERGs (e.g., *Npas4, Arc*) is known to rapidly follow neuronal activation [3]. Indeed, we found induction of these genes in both brain regions 30 minutes after CFC, with higher induction in the mPFC (S1A Fig).

Chromatin immunoprecipitation sequencing (ChIP-Seq) for γH2AX, a chromatin marker of DSBs [7], is a sensitive method for identifying DSBs genome-wide [3, 17–20]. We performed γH2AX ChIP-Seq 30 minutes following CFC to measure the formation of DSBs. In the naive hippocampus we observed 136 γH2AX peaks, increasing to 280 γH2AX peaks after CFC, with 125 peaks shared between conditions (S2 Table). In the naive mPFC we observed 120 γH2AX peaks, increasing to 255 γH2AX peaks after CFC, with 102 peaks shared between conditions (S2 Table). Including all peaks called under the naive and CFC conditions, we found 291 γH2AX peaks annotated to 323 genes in hippocampus, and 273 γH2AX peaks annotated to 306 genes in mPFC (Fig 1A; S1B Fig; S2 Table). Consistent with previous studies, γH2AX peaks were located at gene bodies and proportional to gene length, yet often stretching past the 3'-UTR (S1C Fig) [3, 15, 17].

There was a large overlap between γH2AX peaks called in both hippocampus and mPFC, reflecting their shared recruitment during learning (Fig 1A). We utilized clusterProfiler [21] to perform gene ontology (GO) analysis of these 206 γH2AX peak-containing genes and clustering of the top biological processes yielded four unique categories (S1D Fig). The largest cluster

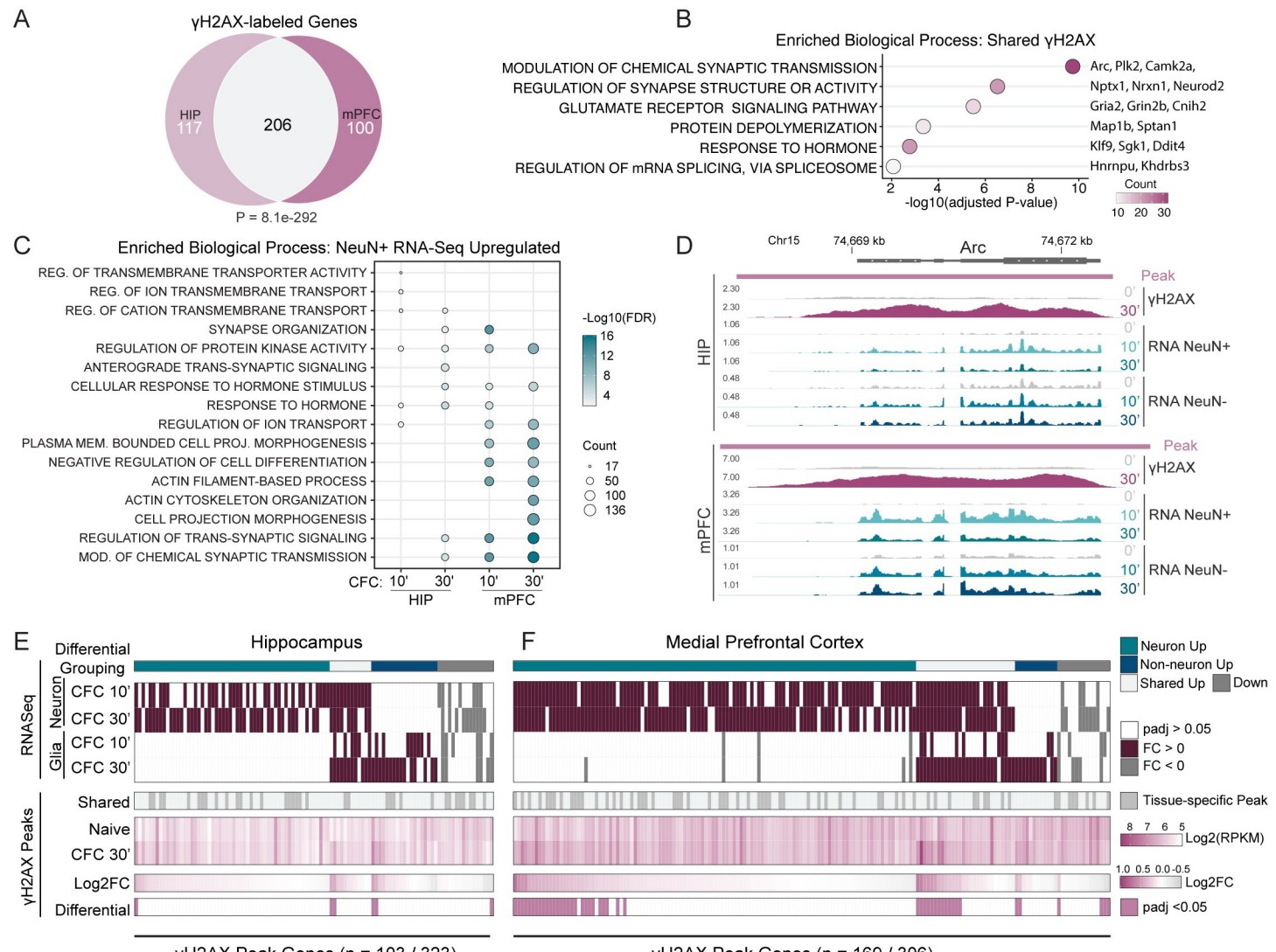

**Fig 1. Fear learning induces DNA double-strand breaks in the brain.** (A) Venn diagram of the γH2AX peak-containing genes shared between HIP and mPFC for both naive and CFC conditions. P-value calculated using hypergeometric distribution test. (B) Six representative top biological processes for the 206 γH2AX peak-containing genes shared between HIP and mPFC in (A). Over-representation analysis with gene ontology (GO) category "Biological Process." (C) The top 5 biological processes for the CFC-upregulated genes in NeuN+ nuclei at each 10- and 30-minute timepoint. Over-representation analysis with gene ontology (GO) category "Biological Process." (D) Genome browser tracks for the gene *Arc*. Both HIP and mPFC are shown. Whole tissue γH2AX ChIP-Seq is shown as LogLR signal tracks ('γH2AX'). Signal normalized total RNA-Seq from FACS-isolated nuclei is shown for neurons ('RNA NeuN+') and non-neurons ('RNA NeuN-'). Time points following contextual fear conditioning are noted; naïve, 10, and 30 minutes (0', 10', 30'). γH2AX ChIP-seq tracks are the combined signal for 3–4 independent replicates, each replicate generated from the pooling of 3 animals. RNA-Seq tracks are the combined signal for 3–4 independent replicates. (E-F) Heatmaps of the genes containing γH2AX peaks that sustained transcriptional regulation after CFC. RNA-Seq heatmap denotes differential genes ('RNA-Seq') and color bar ('Differential Grouping') denotes cell type specificity. The γH2AX heatmaps show peaks shared between tissues ('Shared'), RPKM of γH2AX signal, Log2FC, and those peaks changing after CFC with padj <0.05 ('Differential'). Left is hippocampus (E), Right is mPFC (F).

contained those GOs related to synaptic function (e.g., 'modulation of chemical synaptic transmission') that included glutamate receptors *Gria2* and *Grin2b*, synaptic plasticity regulators like *Camk2a* [22], and ERGs like *Arc* [23] and *Plk2* [24] (Fig 1B). Similar to a previous report in the immune setting, many of these genes are lineage-specific, like the transcription factor *Neurod2* [17]. Two clusters composed of single GO terms were observed, one enriched for RNA binding genes ('Regulation of mRNA splicing, via spliceosome') and one enriched for

cytoskeleton-related genes ('protein depolymerization') (Fig 1B). Finally, the fourth unique cluster was related to hormone or biological rhythms (e.g., 'response to hormone') (Fig 1B). To confirm γH2AX peaks at ERGs, we performed γH2AX ChIP-qPCR on pooled hippocampi collected 30 minutes following CFC. Compared to the naive condition, hippocampi of CFC mice had significant increases in γH2AX at the gene bodies of the ERGs *Npas4* and *Nr4a1*, but not at the housekeeping gene *B2m* (S1E Fig). These findings indicate that many genes essential for neuronal function and memory formation, and significantly more of them than expected based on previous observations in cultured neurons following NMDA stimulation, are potentially hotspots of DSB formation. As DSBs represent a grave threat to genomic integrity [25], with its sequela including transcriptional dysregulation and genomic rearrangements, this suggests that genes critical for neuronal function are uniquely vulnerable to DNA damage.

We previously observed that the formation of DSBs correlated with rapid gene induction in neurons, particularly the ERGs which we find are sites of DSBs in the brain. To understand how these DSBs correlate with CFC-induced gene expression changes, we performed nuclear RNA-Seq. While whole-cell mRNA levels reflect both RNA synthesis and RNA degradation, assaying nuclear RNA levels more directly measures transcriptional activity. We fixed and enriched for neuronal and non-neuronal nuclei collected 10 and 30 minutes after CFC through fluorescence-activated cell sorting (FACS), using the pan-neuronal nuclei marker NeuN [26] (S2A Fig). Nuclei were decrosslinked after sorting and total RNA was isolated for downstream analysis. Utilizing an intronic primer, we found higher transcriptional induction of the neuron-specific ERG *Npas4* [27] in the FACS-isolated neuronal (NeuN+) nuclear RNA than whole mPFC lysate, with minimal expression in the non-neuronal (NeuN-) fraction, indicating successful purification of neurons and non-neurons (S2B Fig). Assaying mRNA of the canonical ERG *Arc* showed induction in both neuronal and non-neuronal nuclei following CFC (S2B Fig). Because the peak of ERG induction occurred as early as 10 minutes or as late as 30 minutes after CFC, we included both time points in our sequencing analyses (S2B Fig).

We next performed nuclear RNA-seq of sorted neurons and non-neurons 10 and 30 minutes subsequent to CFC. First, successful isolation of neuronal nuclei was validated by examining aggregate expression of known cell type-enriched genes [28], finding that pyramidal- and interneuron-enriched genes were highly correlated with the NeuN+ RNA-Seq, while genes enriched in glia, including astrocytes, microglia, and oligodendrocytes, along with other non-neuronal cells were strongly enriched in the NeuN- RNA-Seq (S3A Fig). We identified hundreds of upregulated genes, indicating that fear learning activates the transcriptomes of neurons and non-neurons across brain regions within minutes (S3B–S3E Fig; S3 Table). The mPFC had the highest number of upregulated genes, suggesting a stronger transcriptional response in this area during learning (S3F Fig). In agreement with our γH2AX ChIP-seq analysis, there was a large overlap between HIP and mPFC upregulated genes in neurons (202 genes at 10 minutes and 448 genes at 30 minutes) (S3F Fig). Non-neuronal nuclei also exhibited considerable transcriptional changes in response to CFC, but with more comparable numbers of upregulated genes between brain areas and with a large overlap occurring at 30 minutes (34 genes at 10 minutes and 242 genes at 30 minutes) (S3G Fig). Further, we found biological processes related to synaptic structure and function were amongst the most enriched GO categories in the upregulated genes of neurons–mirroring our γH2AX ChIP-Seq (Fig 1C). In contrast, neuronal downregulated genes had minimal enrichment for biological processes (a single significantly enriched term: "cell-cell adhesion via plasma-membrane adhesion molecules"; adjusted p-value = 2.4x10$^{-3}$).

To assess the relationships between activity-induced DSBs and gene expression in the brain, we compared the ChIP-seq and RNA-seq data. First, examining a specific genomic locus of the ERG Arc revealed increases in γH2AX signal with concomitant upregulation in

both neurons and non-neurons (Fig 1D). Globally, we find four categories of γH2AX-associated genes whose expression was altered after CFC: those upregulated exclusively in neurons (56 HIP and 114 mPFC), genes upregulated in both neurons and non-neurons (12 HIP and 28 mPFC), genes upregulated specifically in non-neurons (19 HIP and 12 mPFC), and a small subset of downregulated genes (16 HIP and 15 mPFC) (categories denoted by "Differential Grouping" row) (Fig 1E and 1F). Overall, we find transcriptional changes are more strongly associated with γH2AX in the brain than anticipated. Previously, we observed twenty gene-associated γH2AX loci following stimulation of cultured neurons [3], while in the HIP and mPFC we see more than 100–150 gene-associated γH2AX loci that are transcriptionally induced (Fig 1E and 1F).

## Activity-dependent genes are a source of DNA breaks in the brain

We next sought to understand the overlap between CFC-upregulated genes and γH2AX peaks. Overall, we found that γH2AX peaks were over-represented with genes upregulated by fear learning, particularly in the mPFC where we saw higher induction of gene expression (Fig 2A and S4A Fig). However, absolute transcription level is known to correlate with DSBs in both human and mouse cells [29–31]. By binning all expressed genes at the CFC30' time point in

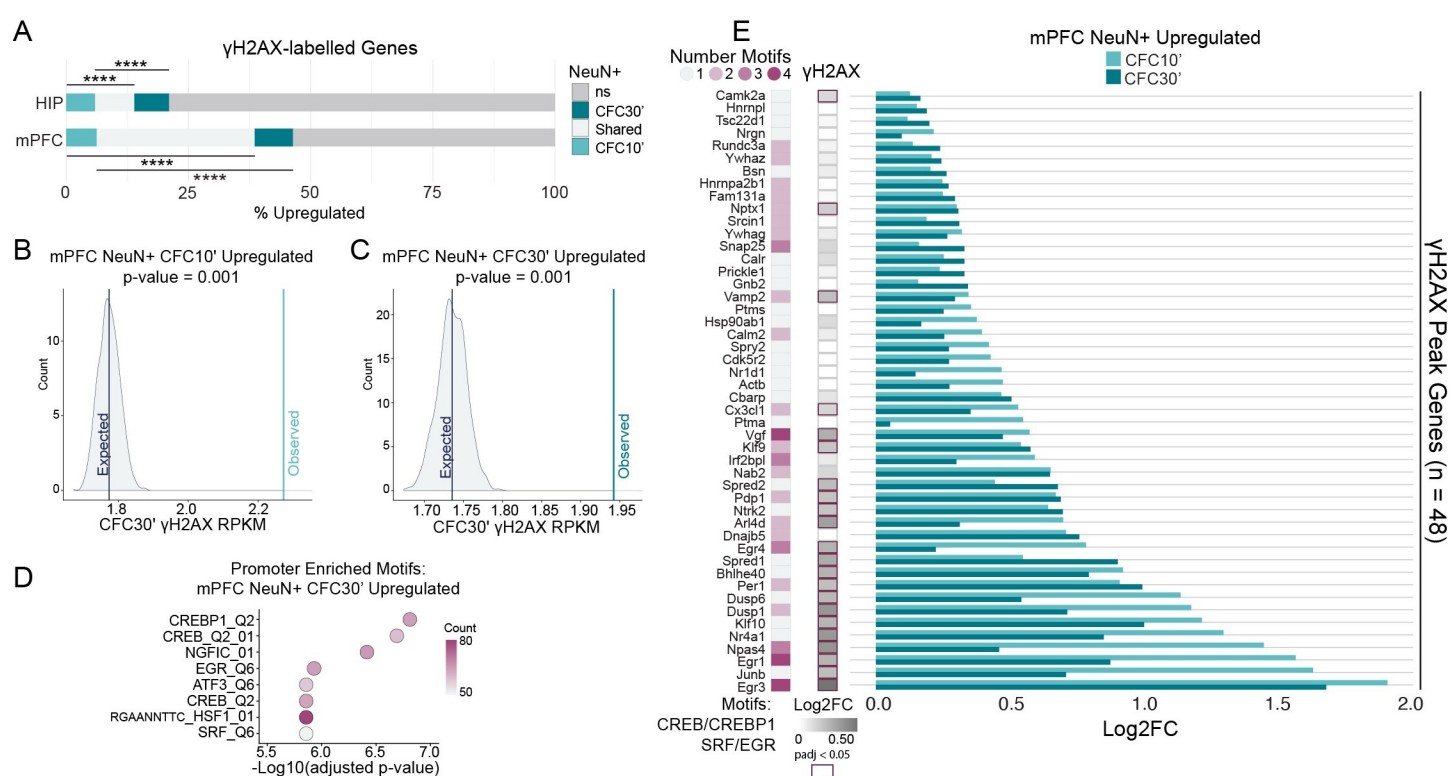

**Fig 2. Activity-dependent genes are a source of DNA breaks in the brain.** (A) Percent overlap between genes containing a γH2AX peak and those that were upregulated (padj < 0.05) in neuronal nuclei after CFC. Hypergeometric distribution test; **** P <0.0001. (B-C) Permutation testing to assess whether CFC-upregulated genes at 10 minutes (B) or 30 minutes (C) have greater than expected γH2AX intensity, accounting for RNA expression level at the same time point (CFC30'). Distributions show the mean γH2AX intensity (RPKM) for 1000 permutations of random sampling, binned by RNA expression level (FPKM). Lines are the mean γH2AX RPKM of either all permutations ('Expected'), or genes upregulated at the specified time point ('Observed'). (D) Top 8 enriched promoter motifs for the genes upregulated in neuronal nuclei from mPFC 30 minutes after CFC (Log2FC >0 & FDR < 0.05). Using the "Transcription Factor Targets" (TFT) gene set from the molecular signatures database (MSigDB). (E) Select activity-associated motifs at the promoters of 48 upregulated genes with DSBs. Left, number of the specified motifs associated with each gene's promoter. Center, γH2AX Log2FC, right, fold change observed after CFC in mPFC NeuN+ nuclei. Using the TFT gene sets from MSigDB for each transcription factor motif.

the mPFC by expression level, we observed that genes with higher RNA expression had higher γH2AX levels in the gene body (S4B Fig). This potentially explains some of the ~55% of the γH2AX-associated genes in the mPFC and ~80% in the HIP that are non-responsive to CFC (Fig 2A).

We next asked whether upregulated genes have higher amounts of γH2AX than can be explained simply by their transcription level alone. Using permutation testing, we binned neuronal upregulated genes by RNA expression level, and these bins were then used for random sampling without replacement from all expressed genes. We found upregulated genes have higher γH2AX intensity than would be expected by their transcriptional level (Fig 2B and 2C and S4C Fig). Further, the more rapid the induction (CFC 10 minutes) the greater the discrepancy between the observed and the expected γH2AX level (Fig 2B and 2C). Thus, while many of the observed sites of DSBs may reflect high expression levels, as exemplified by non-induced highly expressed housekeeping genes like histone genes or neuronal lineage genes (S2 Table), gene induction also appears to correlate with increased γH2AX.

To understand what pathways were mediating the rapid induction of gene expression following CFC in neurons, we searched for transcription factor motif overrepresentation at the promoters of differentially expressed genes, using the Molecular Signatures Database (MSigDB) [32]. Motifs from CREB/ATF family members, EGR family members, as well as SRF, were all enriched (Fig 2D and S5A–S5C Fig). These transcription factors are known to act downstream of cellular activation and calcium influx, including through MAPK signaling [23]. Examining all upregulated genes associated with CREB/ATF, EGR, and SRF motifs for the presence of γH2AX enrichment yields 48 genes in mPFC and 20 genes in HIP (Fig 2F and S5D Fig). Importantly, a number of these activity regulated genes, such as *Npas4*, *Fos*, *Nr4a1*, *Actb*, *Ntrk2*, and *Egr1* are known to be targets of these transcription factors, and are essential for efficient memory formation after CFC [13, 33–37]. Other genes that fit the same category, including *Arc* and *1700016P03Rik* (mir212/mir132) were not included because their regulatory motifs are not in the close vicinity of the TSS [38–40].

Having established a connection between rapid gene induction and γH2AX foci in the brain, we next wanted to understand if any of our DSBs are likely to correspond to late response genes, the second wave of genes induced following stimulation [41]. We compared the observations in the mPFC to a published single-cell RNA-Seq dataset which measured cell-type-specific induction of early-response genes (n = 350) and late-response genes (n = 251) after light stimulation of the visual cortex [42]. We found that the rapidly induced early-response genes are enriched with our mPFC DSB-labeled genes (S6 Fig), with *Tuba1a* the only γH2AX site that is exclusively upregulated at the late-response time point (S6 Fig). This suggests that we are not missing DSBs that occurred at late response genes and recapitulates our nuclear RNA-Seq findings with single cell mRNA. Altogether, these results indicate that DSB formation is more widespread in the brain than previously documented and is associated with an important subset of transcriptionally upregulated genes following CFC.

## Fear learning induces a proteostasis response in neurons and non-neurons

We observed a number of γH2AX-associated genes whose expression was altered after CFC in both neurons and non-neurons (Fig 1E). These included early genes (e.g., *Arc*, *Egr1*) and chaperones (e.g., *Hsp90ab1*, *Hspa8*). We had also observed the heat shock transcription factor HSF1, which induces genes in response to protein folding stress [43], enriched in the promoters of neuronal upregulated genes in both HIP and mPFC (Fig 2D and S5A and S5C Fig). Transcription factor motif analysis of the promoters of genes upregulated in non-neuronal nuclei 30 minutes after CFC yielded both HSF1 and the activity-regulated transcription factor

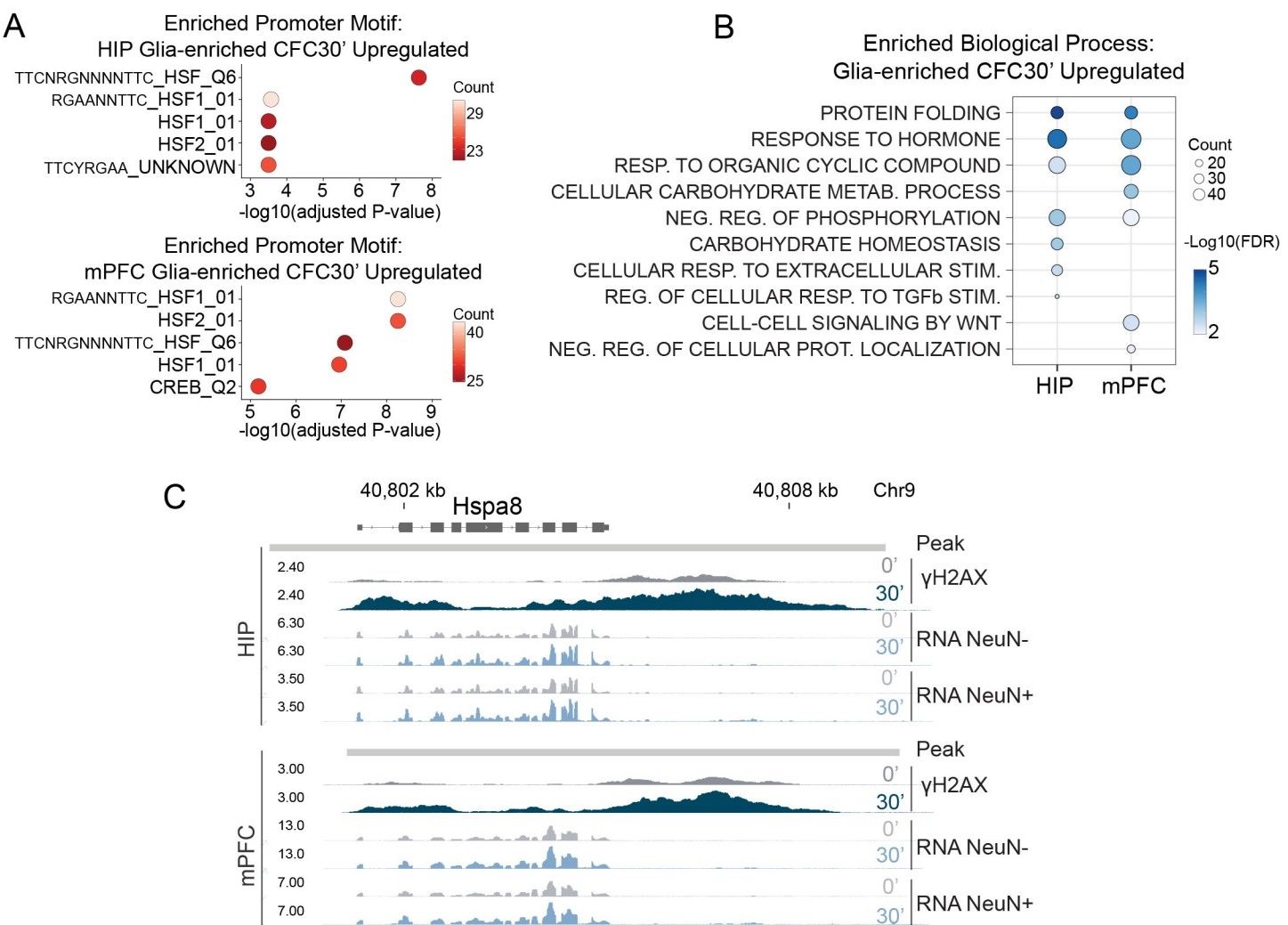

**Fig 3. Fear learning induces a proteostasis response in neurons and non-neurons.** (A) Top 5 enriched promoter motifs at the genes upregulated in HIP NeuN- nuclei (top) and mPFC NeuN- nuclei (bottom) 30 minutes after CFC. Using the TFT gene sets from MSigDB for each transcription factor motif. (B) Ten representative top enriched biological processes for non-neuronal nuclei 30 minutes after CFC. Enrichment for the 426 upregulated genes in HIP and 511 upregulated genes in mPFC. Over-representation analysis with GO category "Biological Process." (C) Genome browser tracks for the chaperone *Hspa8*.

CREB, as in neurons (Fig 3A). Indeed, a number of the CFC-induced genes in non-neuronal nuclei appear to be activity-regulated (S7A Fig). Activation of astrocytes during learning is known to be important for memory formation [44], and these rapid transcriptional responses mediated by activity-regulated transcription factors may reflect an important role of glia in the response to fear learning. We next examined clustering of the top GO terms from these non-neuronal genes and found biological processes related to protein folding, hormone response, metabolism, and signaling (Fig 3B and S7B and S7C Fig). This indicates that CFC elicits a protein folding response and cellular activity-regulated response which is shared by multiple cell types. Inspecting signal tracks for the HSP70 family member *Hspa8* highlighted this relationship, showing the presence of an inducible γH2AX peak (S1B Fig) and increased RNA expression after CFC in both neurons and non-neurons (Fig 3C). Confirming increased HSF1 activity following CFC, we found increased nuclear HSF1 in neurons and non-neurons following CFC, and increased binding of HSF1 to the promoters of Hsp90ab1 and Hspa8 (S8A and S8B Fig).

In HIP and mPFC we found multiple genes with γH2AX peaks that were induced after CFC and which are potential HSF1 targets because of promoter HSF1 binding following heat shock in mouse embryonic fibroblasts [45] (HIP: *Hspa8*, *Baiap2*, *Sh3gl1*, *Dnaja1*, *Hsp90ab1*, *Dynll1*, *Mbp*, *Ywhah*, *Dnajb5*, *Ddit4*, *Prkag2*, *Gse1*, *Ptk2b*, *Arpc2*, *Ywhag*; mPFC: *Tcf4*, *Hspa8*, *Baiap2*, *Hsp90ab1*, *Hnrnpa2b1*, *Gfod1*, *Lncpint*, *Ywhah*, *Dnajb5*, *Ddit4*, *Ywhag*). We also identified ATF6, which functions as part of the unfolded protein response (UPR) and facilitates protein quality control in the endoplasmic reticulum [46], as a potential regulator of additional genes. Known ATF6 targets such as *Hspa5* (Grp78) [47], *Calr* [47], *Xbp1* [48], and others (*Ywhaz*, *Atp2b1*) [49], were enriched with γH2AX peaks and upregulated in neurons and to a lesser degree non-neurons. These findings indicate that CFC generates a rapid proteostasis response in both neurons and non-neurons, with induced genes constituting sites of DNA breaks.

## Glucocorticoid-regulated genes are sites of DNA double-strand breaks

'Response to hormone' was one of the top enriched biological processes observed amongst the CFC-induced genes in non-neuronal nuclei (Fig 3B) as well as the γH2AX peaks (Fig 1B). Examining these genes further, we found examples such as *Sgk1* and *Ddit4* which are known to be regulated by the glucocorticoid receptor (GR) [50, 51] and while not upregulated in neuronal nuclei, were upregulated at the mRNA level in whole HIP and mPFC lysate (Fig 4A and S9A Fig). Unlike neuronal activity, which occurs immediately upon exposure to environmental changes, the hormonal response to stress is delayed while the signal is relayed through the hypothalamic-pituitary-adrenal axis, before eliciting glucocorticoid release into the blood stream. Glucocorticoids increase in the blood within 30 minutes following exposure to a stressor [52], corresponding with increases in the intrahippocampal corticosterone concentration [53] and nuclear localization of the GR in the mouse brain [52]. We observed that 30 minutes was the time point where non-neuronal CFC-upregulated genes were most likely associated with a γH2AX peak (Fig 4A). Furthermore, compared to other brain areas, the mPFC and HIP have some of the highest expression of GR [54], suggesting they are key targets of the stress response. To identify putative GR-regulated genes, we utilized two ChIP-Seq datasets of GR binding in rat cortex to map all binding sites containing the glucocorticoid-responsive element (GRE) in the mouse genome to the nearest gene (S4 Table) [55, 56]. Interestingly, we found that many of the γH2AX-containing genes that were responsive to CFC only in non-neuronal nuclei are coincident with genes annotated to a GR-binding site (Fig 4A).

We tested whether a subset of these genes can be induced by the GR-specific agonist dexamethasone in cultured primary glia. In contrast to *Actb* which is not a known target of GR, we found dexamethasone induced the expression of *Ddit4*, *Sgk1*, and *Glul*, genes that were specifically upregulated in non-neuronal nuclei during CFC and annotated to a GR-binding site (Fig 4B). Thus, our findings implicated the GR in mediating gene induction in glia after fear learning. Next, to assess whether GR activity is sufficient to increase DSBs at these genes, we treated cultured primary glia with dexamethasone and measured γH2AX enrichment by ChIP-qPCR. The genes *Ddit4*, *Glul*, and *Sgk1*, alongside the canonical GR-inducible gene *Mt1* [58], showed significant increases in γH2AX enrichment (Fig 4C). *Arc*, with similarly high γH2AX levels following CFC, alongside the housekeeping gene *B2m*, did not exhibit γH2AX enrichment in response to dexamethasone (S9B Fig).

Our RNA-seq data from sorted nuclei showed upregulation of the γH2AX-associated gene *Ddit4* only in non-neuronal nuclei following CFC, a similar pattern for many of our other putative and confirmed GR-regulated genes (Fig 4D). To understand whether non-neurons had more active GR-bound enhancers, we utilized a ChIP-Seq dataset of histone 3 lysine 27

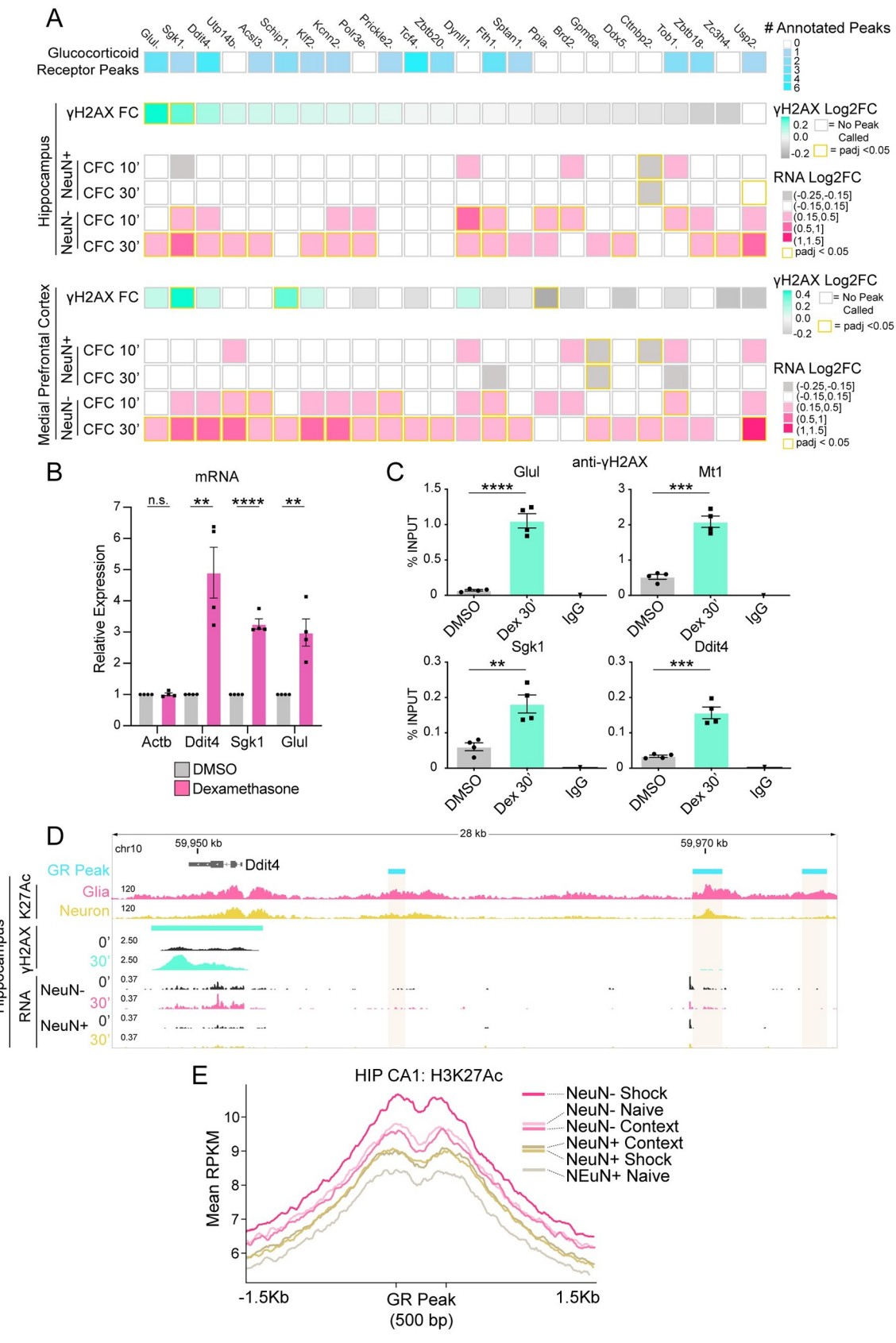

**Fig 4. Glucocorticoid-regulated genes are sites of DNA double-strand breaks.** (A) Heatmap of γH2AX peaks occurring at genes upregulated specifically in non-neuronal nuclei. Top, number of glucocorticoid receptor binding sites annotated per gene (rat cortical ChIP-Seq) [55, 56]. γH2AX Log2FC and upregulated genes for HIP and mPFC after CFC. (B) RT-qPCR analysis of mRNA induction in mouse glial primary cultures 2 hours after treatment with glucocorticoid receptor agonist dexamethasone (100nM). N = 4 independent cultures; two-tailed unpaired student's t-test; ** P ≤ 0.01; *** P ≤ 0.001; **** P <0.0001; Mean ± SEM. (C) ChIP-qPCR analysis of γH2AX induction at select gene bodies in mouse glial primary cultures 30 minutes after treatment with dexamethasone (Dex) (100nM). N = 4 independent cultures; two-tailed unpaired student's t-test; ** P ≤ 0.01; *** P ≤ 0.001; **** P <0.0001; Mean ± SEM. (D) Genome browser snapshot of the gene *Ddit4*. Top, glucocorticoid receptor binding sites ('GR Peak'; rat cortical ChIP-Seq) [55, 56], hippocampal region CA1 H3K27ac ChIP-Seq from NeuN+ or NeuN- isolated nuclei 1 Hour after CFC ('K27Ac') [57], γH2AX LogLR signal tracks, and nuclear RNA-Seq. (E) Average H3K27ac signal at glucocorticoid receptor binding sites (rat cortical ChIP-Seq) [55, 56] containing the GR motif in mouse (n = 5591 peaks). H3K27ac ChIP-Seq of mouse hippocampal CA1 region from NeuN+ or NeuN- isolated nuclei 1 hour after exposure to context, or CFC ('shock') [57]. Colored bars represent the apex of each condition.

acetylation (H3K27Ac), a chromatin mark of enhancer and promoter activity, from purified neuronal and non-neuronal nuclei [57]. Glia have higher H3K27Ac signal at the GR-bound enhancers surrounding *Ddit4*, indicating that GR-regulated enhancers are more active in non-neuronal nuclei than neurons (Fig 4D). To examine this phenomenon genome-wide, we looked at aggregate H3K27Ac signal in neurons and non-neurons at all GR-binding sites (S4 Table) [55, 56]. Strikingly, both the anterior cingulate cortex (ACC) of the medial prefrontal cortex, and the hippocampal area Cornu Ammonis 1 (CA1) showed higher baseline acetylation around GR peaks in non-neurons vs. neurons ('Naive') (Fig 4E and S9C Fig). We then examined H3K27Ac signal in CA1 under additional experimental conditions including 'context' (exposure to the context without a foot shock) and 'shock' (context paired with a foot shock). We found that H3K27Ac signal at GR peaks in the neuronal fraction increased similarly after exposure to either context or shock, suggesting a generalized enhancer activation in response to exploratory behavior that may be independent of stress. In contrast, the non-neuronal fraction showed increases in H3K27Ac after shock, demonstrating that these enhancers are responsive to the stressful condition in non-neurons but not in neurons (Fig 4E; S9D Fig, intergenic peaks).

Our findings identified a group of CFC-responsive non-neuronal genes that are likely regulated by GR signaling (Fig 4A–4E). We checked gene expression of the GR gene, *Nr3c1*, finding that neurons express *Nr3c1* at approximately half the level of non-neurons (S9E Fig). The differing GR expression levels could be one of the reasons why these same genes did not exhibit induction or increased enhancer activity in neurons (Fig 4A). Therefore, we verified whether GR nuclear translocation occurs in response to receptor agonism in both cell types. We measured GR nuclear intensity in mouse brain after treatment with corticosterone, the predominant glucocorticoid in rodents [59]. We found increases in nuclear GR in both neurons and non-neurons (though there was a trend, it was not significant in the NeuN- fraction) (S9F Fig). Thus, the absence of a neuronal stress-mediated change in enhancer activity is likely due to decreased chromatin accessibility at the enhancer level [60], highlighting that glia may play a significant role in the homeostatic response to stress. Nevertheless, it is unclear whether neurons are capable of mounting a transcriptional response to stress hormone, and whether induction of hormone-responsive genes in neurons would be accompanied by DSBs.

## Glia but not neurons have a robust transcriptional response to corticosterone

To test whether an endogenous GR agonist was sufficient to upregulate some of the glial genes displaying elevated levels of γH2AX and transcription following CFC, we injected mice with corticosterone at a dose known to approximate a stressful experience [61], and collected the hippocampus 30 minutes later. We FACS-sorted nuclei into four cell populations: neuronal

(NeuN+), astrocytic (GFAP+), microglial (PU.1+), and oligodendrocyte-enriched (NeuN-, GFAP-, PU.1-; 3X-), and subjected them to RNA extraction (S10A Fig). RT-qPCR analysis showed enrichment for respective cell type markers, indicating successful isolation of cell types (S10B Fig). We then assessed gene expression changes in *Sgk1* and *Glul* that have CFC-inducible γH2AX peaks (S2A Fig), and found that except for neurons, all three glial subtypes could respond to an endogenous GR agonist (S10C Fig). While GR agonists are sufficient to induce the putative glucocorticoid-regulated genes after CFC both *in vitro* (dexamethasone; Fig 4B) and *in vivo* (corticosterone; S10C Fig), we sought to determine whether these genes are dependent on the GR for CFC-induced changes in expression. We found that whereas pretreatment with a glucocorticoid receptor antagonist RU-486 (mifepristone) [62] blocked CFC-induced transcription of *Sgk1*, *Ddit4*, and *Glul* in whole hippocampal lysates, it did not alter transcription of the housekeeping gene *Gapdh*, or induction of the ERG *Arc* (Fig 5A).

We next performed RNA-Seq from hippocampal cell types after corticosterone treatment to better understand how the transcriptomes of the four major brain cell types respond to GR-mediated transcriptional regulation. Successful isolation of brain cell types was validated by examining aggregate expression of known cell type-enriched genes [28] (S11A Fig). Neurons have a modest transcriptional response following corticosterone treatment (112 genes; Fig 5B and S11B Fig; S3 Table). In contrast, astrocytes, oligodendrocyte-enriched, and microglia have hundreds of upregulated genes (276, 453, and 551 respectively; Fig 5B and S11C–S11E Fig; S3 Table). Our results are consistent with published *in vitro* findings that reported extensive response to dexamethasone in cultured astrocytes but little in cultured neurons [63]. The ability of glia to mount a robust transcriptional response to glucocorticoids suggests that glia may have a much larger role to play in the response to stress and its impact on the brain during learning than previously appreciated.

Clustering of the top GO terms from the genes upregulated following corticosterone treatment shows major categories of biological processes pertaining to proliferation, cell death, cellular motility, homeostasis, signaling, inflammation, other various cellular functions, and as would be expected, a glucocorticoid response (Fig 5C; S12A–S12C Fig). No enriched terms were observed within the neuronal upregulated genes. Downregulated genes were enriched for biological processes related to cell motility, inflammation, differentiation and proliferation (S13A–S13D Fig). Glial function is known to be affected by cellular activity and motility, with morphological changes reflecting changes in cellular function [64–66]. Together, these large changes in the transcriptomes of the three glial cell types is likely to impact their functions and could affect the formation of memory.

We next sought to understand how well GR-mediated gene induction could explain the glia-specific DSBs seen *in vivo*, and whether genes regulated through this pathway in neurons incur DSBs. Examining all γH2AX-containing genes that were also upregulated in one of the cell types after corticosterone, we found that the vast majority (32/43; 74%) are regulated only in glia (Fig 5D). Thus, we have identified a glial-enriched pathway that may be incurring DSBs during CFC. Collectively, these results show that genes responsive to stress hormone are predominantly glial, with some of these genes showing high levels of the DSB marker γH2AX and likely modulating important glial functions.

## Discussion

There is increasing evidence for an association between neuronal activity and the generation of DSBs, but their *in vivo* location and relation to brain function is unknown [1–3, 5, 6]. Here, using γH2AX as a proxy for DSBs, we identify hundreds of gene-associated DSBs in the medial

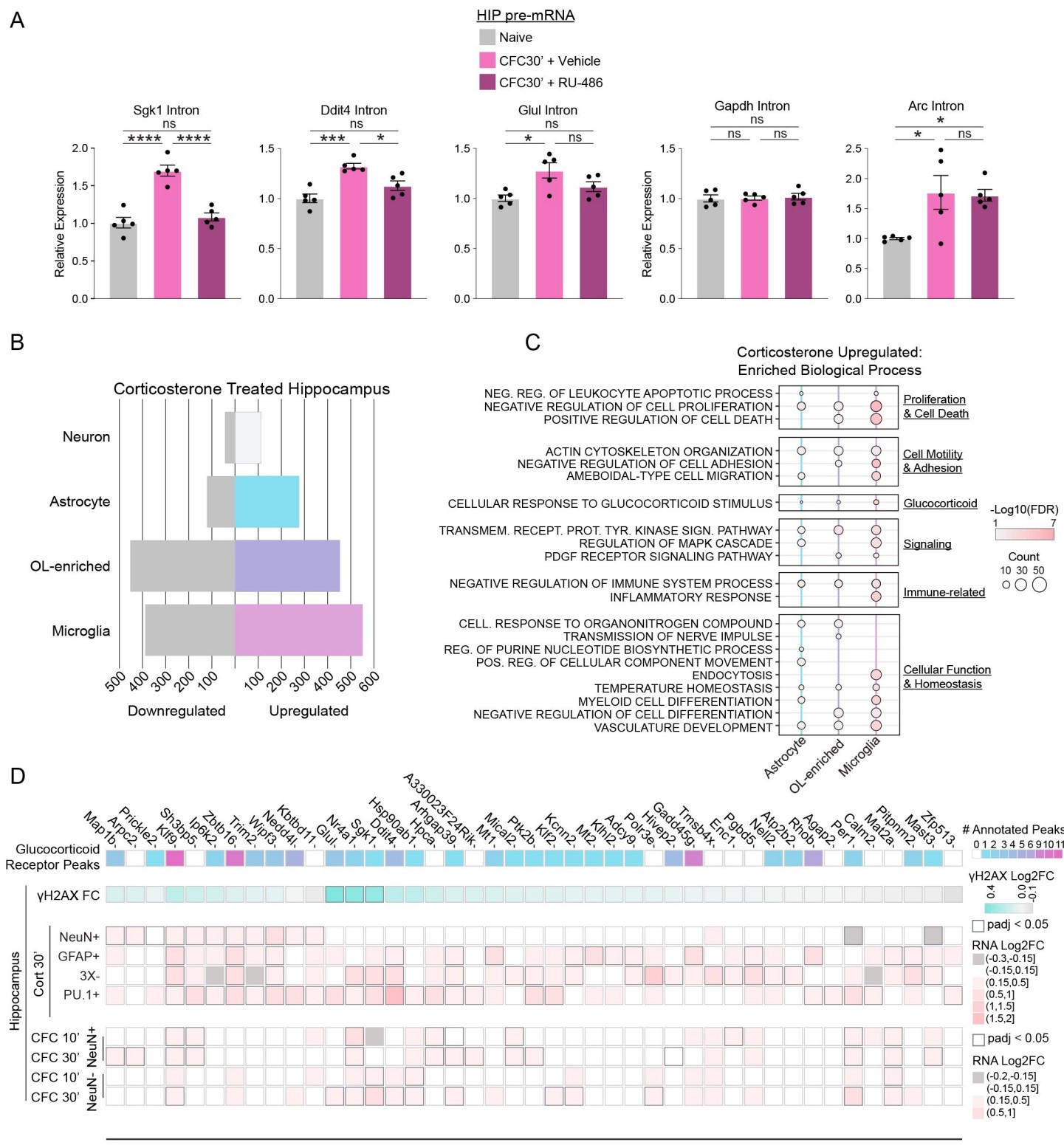

**Fig 5. Glia, not neurons, have a robust transcriptional response to corticosterone.** (A) Pretreatment with glucocorticoid receptor antagonist RU-486 (Mifepristone) blocks CFC-induced gene expression in hippocampus. Pretreatment with vehicle (1% v/v Tween 80 in saline) or 50 mg/kg RU-486 IP occurred 30 minutes prior to CFC. qRT-PCR analysis of pre-mRNA with intronic primer and normalized to *Hprt*. cDNA was primed with random hexamers. N = 5 mice per group; one-way ANOVA with Tukey's multiple comparisons test. (B) Number of corticosterone upregulated and downregulated genes from RNA-Seq of FACS-isolated nuclei from hippocampal cell

types 30 minutes after saline or corticosterone:HBC complex (2mg/Kg) treatment. Cutoff padj < 0.05. (C) Twenty-one representative top enriched biological processes for the upregulated genes in purified hippocampal nuclei from astrocytes, microglia, and oligodendrocyte-enriched after corticosterone treatment (276, 453, and 551 genes respectively; padj < 0.05). Summary categories representing each grouped list of GOs is listed on the right. No enrichment of processes at threshold padj < 0.05 with the 112 upregulated genes in neuronal nuclei. Over-representation analysis with gene ontology (GO) category "Biological Process." (D) Heatmap of 43 γH2AX peaks at genes upregulated in FACS-isolated neurons and glia following corticosterone treatment. From top: number of glucocorticoid receptor (GC) binding sites annotated per gene (rat cortical ChIP-Seq) [55, 56], γH2AX Log2FC in HIP after CFC, corticosterone-induced genes in NeuN+, GFAP+, PU.1+, and oligodendrocyte-enriched (3X-; NeuN-GFAP-PU.1-) hippocampal nuclear RNA-Seq 30 minutes after corticosterone treatment, and HIP RNA-Seq after CFC.

prefrontal cortex and hippocampus that are important for learning and memory [16]. The surprisingly high number of genes with DSBs expands upon the small number previously observed in neurons following NMDA stimulation *in vitro* [3].

We observed that gene induction exhibits higher γH2AX than expected based on gene expression level, and disparate classes of γH2AX peaks, such as lincRNAs (*Lncpint*, *Mir9-3hg*, *Mir9-1hg*, *1700016P03Rik*[mir212/mir132]), housekeeping genes (*Hsp90ab1*, *Actb*), and as seen previously, lineage-specific genes [17], particularly those related to neuronal function (*Grin2b*, *Camk2a*, *Cck*, *Mbp*), are all regulated by CFC. However, though there is a clear correspondence between γH2AX peaks and CFC-induced genes, we do see less significant changes in γH2AX enrichment at many of these genes, with most of the peaks already present in the naive condition. In our previous study, significant γH2AX peaks become evident only after inducing neuronal activity in cultured neurons [3], suggesting that the presence of DSBs in the naive condition at known activity-induced genes may partially reflect basal neuronal activation occurring in the brain.

γH2AX is a sensitive marker of DSBs [67] that has been successfully used to map sites of DSBs occurring genome-wide through ChIP-Seq profiling [3, 17], including at known cleavage sites following the induction of restriction enzymes [18, 19, 68]. However, in some instances γH2AX is not reflective of underlying DSBs [69], including in neurons where pan-nuclear γH2AX staining was observed without evidence of DSBs during non-physiologic stimulation [70]. Additionally, two closely spaced single-strand DNA breaks (SSBs) could potentially be recognized as a DSB by the cell [71]. γH2AX increases following induction of SSBs in postmitotic neurons were dependent on transcription [72], suggesting that SSBs may require conversion to a DSB before generating a DNA damage response (DDR) and γH2AX formation [73]. While γH2AX ChIP-Seq has the benefit of measuring the DDR, profiling those DSBs in the genome that the cell recognizes as dangerous and in need of repair, it is important that methods which measure DNA breaks directly are utilized to both confirm the current results and to extend them through cell-type-specific analysis.

As most brain cells are postmitotic, they rely on non-homologous end joining (NHEJ) for DNA double-strand break (DSB) repair [25]. NHEJ can be error free, however, the presence of blocked DNA ends promotes end resection, which can result in sequence loss, rearrangements, or translocations [74]. The accumulation of irreversible sequence damage with time has the potential to perturb brain function during aging and disease [9], and efficient DNA repair pathways are thought to be critical to prevent functional decline during brain aging and neurodegeneration [25, 75]. ERGs and heat shock genes, two classes of DSB hotspots that were induced following CFC in neurons and non-neurons, were found in the aged pancreas to be sites of transcriptional noise and this correlated with the presence of somatic mutations [76]. It is interesting to speculate whether the same process also occurs in the brain with age and whether it may compromise the brain's ability to respond to cellular insults occurring during aging or in neurodegeneration, where protein folding factors are upregulated during disease progression [77]. Whether their overexpression contributes to the accumulation of DNA breaks observed during the progression of neurodegenerative disease is unclear [70, 78]. Overall, we have identified sites of DSBs at genes important for neuronal and glial functions,

suggesting that impaired DNA repair of these recurrent DNA breaks which are generated as part of brain activity could result in genomic instability that contribute to aging and disease in the brain [9, 25].

Convergent transcription that leads to polymerase collision is known to generate DSBs [17, 79]. We observed a few instances in which small γH2AX peaks are found near sites of antisense transcription. For example, a small γH2AX peak is present within intron 1 of *Polr3e*, a known site of transcriptional interference between RNA polymerase II (Pol II) and antisense transcription mediated by RNA polymerase III (Pol III) (S14 Fig) [80]. Other examples include a small peak at *29000060B14Rik* that is within and antisense to *Clasp1*, the promoter of *Pcif1*, or other peaks which overlap the 3' UTRs of closely spaced genes (e.g., *Prrc2a*/*Bag6*, *Dbn1*/*Prr7*).

We found that glia are likely to play an underappreciated role in the response to stress in the nervous system and this corresponds with DSBs, a relation between stress hormones and DNA damage that was also observed in mouse fibroblasts [81]. Our results are reminiscent of observations in the nucleus accumbens after morphine treatment, where oligodendrocytes in particular were found to induce a number of genes targeted by the GR [82]. Why neurons exhibit such limited responses to corticosterone remains uncertain. However, given our observation that GR-bound enhancers are more active in glia, and that GR nuclear intensity increased in neurons after corticosterone treatment, it is likely that chromatin accessibility plays a key role determining the GR response, as reported previously [60]. This indicates a predominately epigenetic mechanism underpinning the modest transcriptional response that neurons exhibit to corticosterone.

We find that stress likely impacts the physiology of glia through modulation of their transcriptomes, impacting numerous cellular processes. These changes may explain how stress has been shown to impact glial morphology and function, including after CFC [50, 83–91]. The role of glucocorticoids during the brain's response to stress could therefore be partially separated into a predominantly non-transcriptional role in neurons, wherein the GR has an important transcription-independent function at the synapse that aids memory formation [59, 92]. In contrast, the homeostatic response to stress may run primarily through glia, consistent with the general role of glia in brain homeostasis. Beyond homeostasis, astrocytic GR expression was found to be necessary for CFC-induced memory formation [91], and future work will be required to better understand how glia facilitate or hinder learning through their GR response.

Our observations suggest that the glial contribution to the deleterious effects of stress hormones may be stronger than previously appreciated. This may include cases of steroid dementia, wherein cognitive alterations occur in response to high levels of glucocorticoids (e.g., Cushing's syndrome), as well as disorders characterized by anxiety and depression [93, 94]. Interestingly, the microglial gene expression signature seen after corticosterone treatment was enriched for disease associations such as inflammation and depression (S15 Fig). This fits with the observation that stress can potentiate the microglial inflammatory responses [95–97], and their implication in the etiology of depression [98]. Because we found that glial GR-bound enhancers are more active in responding to stress than those in neurons, we posit that susceptibility to stress may include an underappreciated genetic component comprised of glia-specific variants. This also implicates glia, particularly microglia, in the genetics of the many psychiatric and neurodegenerative disorders for which stress is a risk factor [99–102], including Alzheimer's disease [103, 104] and schizophrenia [105].

## Materials and methods

### Ethics statement

All mouse work was approved by the Committee for Animal Care of the Division of Comparative Medicine at Massachusetts Institute of Technology (protocol number 0618-044-21).

## Contextual fear conditioning paradigm, treatments, and tissue collection

Sixty-eight 4-month-old C57BL6/J male mice were purchased from Jackson Laboratory (stock number 000664). Mice were group housed with a 12-hour light and dark cycle with access to food and water ad libitum. To minimize variability, mice were single housed for one week before experimental manipulation.

For contextual fear conditioning, mice were habituated in the context for 3 minutes prior to administration of 30 second-spaced dual 0.8 mA foot shocks applied by the grid floor. The animals remained in the chamber for an additional minute and were placed back in their home-cage. Ten or thirty minutes after placement in the context, mice were euthanized by cervical dislocation. For Mifepristone (Sigma-Aldrich) pretreatment, mifepristone was dissolved in 1% v/v Tween 80 in saline. Mice were treated IP with either 50mg/Kg mifepristone or vehicle immediately before contextual fear conditioning, followed by euthanasia 30 minutes later. For corticosterone treatment, corticosterone:HBC complex (Sigma-Aldrich) was dissolved in saline and administered at 2mg/Kg IP, or an equal volume of saline for control, followed by euthanasia 30 minutes later.

Treatment and control groups were euthanized in a staggered manner to minimize circadian differences between groups. Naive mice remained in their home cages prior to euthanasia. For tissue collection, the animals were sacrificed by cervical dislocation and the brains were rapidly extracted and submerged in ice-cold PBS. To isolate the medial prefrontal cortex and hippocampus, the brain was placed ventral side up in an Alto coronal 0.5mm mouse matrix resting on ice. Three coronal cuts were administered with razor blades, one separating the PFC from the olfactory bulb, one placed approximately around the optic chiasm to separate the PFC from the hippocampus, and one placed within the cerebellum for stability. The pieces containing the mPFC and hippocampi were placed in an ice-cold PBS-filled dish for isolation with a dissection microscope. To isolate the mPFC, a horizontal cut was administered just above the anterior olfactory nucleus with a razor blade. Two longitudinal cuts were made medial to the anterior forceps of the corpus collosum. Whole hippocampi were unfurled and isolated from within the cortex. White matter and either meninges, or choroid plexus [106], were removed to prevent contamination. Tissue was flash frozen in liquid nitrogen and stored at -80˚C until processing.

## Mixed glial cultures and treatments

For mixed glial cultures, four Swiss-Webster timed-pregnant mice were ordered from Charles River (stock number 024). Cortical glia from pups younger than postnatal day 6 were cultured essentially as described [107]. Briefly, cortices were dissociated with papain (Worthington Biochemical) and plated onto non-coated 10 cM petri dishes. Mixed glia were cultured for a minimum of one week in DMEM containing 10% FBS, GlutaMAX, and Pen-Strep at 37˚C and 5% $CO_2$. For treatment with glucocorticoid receptor agonist, dexamethasone (Sigma-Aldrich) was dissolved in DMSO and applied to the media at 100nM. For ChIP experiments, the cultured cells were fixed by diluting 16% Methanol-free Paraformaldehyde (Electron Microscopy Sciences) to 1% in the culturing media and rocked for 10 minutes, before quenching with 0.25M Tris pH 8 (a more effective quencher than glycine [108]). Cells were then scraped, and nuclei were released by resuspending in NF1 buffer (0.5% Triton X-100, 0.1M Sucrose, 5mM MgCl2, 1mM EDTA) and dounce-homogenized with a loose pestle for 30 strokes. Nuclei were centrifuged for 15 minutes 2000 RCF 4˚C and the supernatant aspirated, leaving nuclei for downstream applications.

## Tissue homogenization

Tissue was dissociated with a motorized pestle (Argos Technologies) in 0.3–0.5mL ice cold PBS treated with protease and phosphatase inhibitors (cOmplete & PhosSTOP; Roche). For

RNA isolation, RNAse inhibitors were added to all buffers (RiboLock (Thermo Scientific) or SUPERaseIn (Invitrogen); 1:100 for homogenization, 1:1000 for buffers with BSA, and 1:10,000 for other buffers). Dissociated brain tissue was fixed in 10 mL of 1% Methanol-free Paraformaldehyde (16%; Electron Microscopy Sciences) for 10 minutes before quenching with 0.25M Tris-HCl pH 8 (a more effective quencher than glycine [108]). Homogenate was centrifuged for 15 minutes 2000 RCF 4˚C and the supernatant aspirated. Nuclei were released by resuspending in NF1 buffer (0.5% Triton X-100, 0.1M Sucrose, 5mM MgCl2, 1mM EDTA) and dounce-homogenized with a loose pestle for 30 strokes, then filtered with 70uM cell strainers (Falcon). Nuclei were centrifuged for 15 minutes 2000 RCF 4˚C and the supernatant aspirated, leaving nuclei for downstream applications.

## Whole-cell mRNA processing

Extraction of mRNA from whole tissue and cultured mixed glia was performed with the RNeasy mini kit (Qiagen). For brain tissue, homogenization was performed by aspirating the tissue in RLT Plus buffer through a 20-gauge needle and syringe approximately 10 times until homogenized. For cell culture, the media was aspirated before RLT Plus was added and distributed with rocking. Purification proceeded as described by the manufacturer. Isolated RNA was quantified on a NanoDrop spectrophotometer (Thermo Fisher Scientific) and 1ug RNA was used to make cDNA with the OligodT RNA to cDNA EcoDry Premix (Takara) according to the manufacturer's instructions, before proceeding to qPCR analysis.

## qPCR

For qPCR analysis, diluted cDNA or genomic DNA was subjected to quantitative real-time PCR in triplicate with the indicated primers using Ssofast EvaGreen Supermix (Bio-Rad) in a CFX Connect Real-Time System (Bio-Rad). For gene expression analysis, normalization was against *Hprt* using the ΔΔCT method. For ChIP, normalization was against Input. Primer sequences can be found in S1 Table.

## Flow cytometry

Fixed brain nuclei (see 'Tissue Homogenization') were resuspended in 1mL 0.5% BSA in PBS (IgG-Free, Protease-Free; Jackson ImmunoResearch). Nuclei were stained in Eppendorf tubes with the relevant antibodies rocking for 30–60 minutes at 4˚C. For neuron and glia isolation, nuclei were stained with NeuN AF488 (1:1000; clone MAB377X; Millipore). For neuron and glial subtype isolation, nuclei were stained with NeuN AF488 (1:1000; clone MAB377X; Millipore), GFAP AF647 (1:200; clone GA5; Cell Signaling Technology), and PU.1 PE (1:200; clone 9G7; Cell Signaling Technology). To stain for the glucocorticoid receptor, nuclei were incubated with Polyclonal Glucocorticoid Receptor (2 ug; clone PA1-511A, Thermo Fisher Scientific) followed by Donkey anti-Rabbit IgG AF647 (0.5ug; A-31573; Thermo Fisher Scientific). To stain for HSF1 receptor, nuclei were incubated with anti-HSF1 (1:250; 4356S; Cell Signaling Technologies) followed by Donkey anti-Rabbit IgG AF647 (0.5ug; A-31573; Thermo Fisher Scientific). Nuclei were pelleted between steps by centrifugation for 10–15 minutes at 2000RCF at 4˚C. Finally, to help gate for singlet nuclei, 1:1000 DAPI (Sigma-Aldrich) was added to the buffer just prior to flow cytometry. Nuclei were then run on a LSRII cytometer (BD Biosciences) or isolated with a BD FACSAria (BD Biosciences) cell sorter into 1% BSA PBS with inhibitors. The data was analyzed with FlowJo software (FlowJo LLC).

## Nuclear RNA isolation, cDNA generation, and sequencing

FACS-isolated nuclei (see 'Flow Cytometry') were pelleted by centrifugation for 15 minutes at 2000RCF at 4˚C. To decrosslink the nuclei, the RecoverAll Total Nucleic Acid Isolation Kit for FFPE (Thermo Fisher Scientific) was utilized following the manufacturer's instructions. Briefly, nuclei were resuspended in 200uL digestion buffer with 4 uL protease (an equal volume of protease K (NEB) was substituted if the manufacturer-provided protease was exhausted) for 15 minutes at 50C, then 15 minutes at 80C. To isolate the RNA and eliminate most DNA prior to DNAase treatment, 800uL TRIzol LS (Invitrogen) was added, mixed well, and incubated for 5 minutes at room temperature before proceeding with isolation or freezing at -80C. 215uL chloroform was added to the solution and vortexed vigorously for 30 seconds before adding to a 5Prime Phase Lock Gel Heavy tube (Quantabio) and centrifuged for 15 minutes at 12,000g at 4C before transfer to an eppendorf tube. An equal volume of 100% ethanol (800uL) was added immediately and mixed well before proceeding to RNA isolation with the Direct-zol RNA Microprep Kit (Zymo Research). DNase treatment and RNA isolation proceeded according to the manufacturer's instructions, before elution in 6-20uL water.

The generation of cDNA from the isolated nuclear RNA was performed with SuperScript III or IV (Invitrogen) according to the manufacturer's instructions, priming with either random hexamer or oligo(dT) primers. Diluted cDNA was then utilized for qPCR (see 'qPCR').

For library preparation, RNA concentration and quality was assessed with a Fragment Analyzer (Agilent), yielding an RNA fragment distribution concentrated between approximately 200bp to 6000bp. RNA-Seq libraries were generated with the SMARTer Stranded Total RNA-Seq Kit—Pico Input Mammalian (v1 or v2; Takara) according to the manufacturer's instructions. Because the RNA was already partially degraded during the fixation and decrosslinking procedure, the RNA fragmentation time was 90 seconds. Paired end sequencing was performed with a NextSeq500 at the MIT BioMicro Center.

## ChIP and ChIP-seq

Performed similar to [109]. Pelleted nuclei from cultured mixed glia (see 'Mixed Glial Cultures and Treatments') or dissociated brain tissue combined from three different animals' hippocampi or mPFC (see 'Tissue Homogenization'), were lysed by the addition of 400 μl LB3 (1mM EDTA pH 8, 0.5mM EGTA pH 8, 10 mM Tris pH 8, 0.5% Sarkolsyl solution) and split into 2 tubes and sonicated on 'HIGH' for 30–40 cycles (30" on and 30" off) in a Bioruptor bath sonicator (Diagenode). The immunoprecipitation was prepared by diluting 15-30ug of the chromatin into 1% Triton X-100, 0.1% sodium deoxycholate, 1 mM EDTA plus protease and phosphatase tablets (cOmplete & PhosSTOP; Roche) and preclearing with Protein A Dynabeads (Life Technologies) blocked with BSA. Then 5 ug anti-γH2AX (ab2893; Abcam) or 10 uL anti-HSF1 (clone 4356S; Cell Signaling Technologies) were added and the chromatin rotated overnight. BSA blocked Protein A Dynabeads (Life Technologies) were added and rocked for 4 hours before 4 washes with RIPA buffer (50 mM HEPES, pH 7.6, 10 mM EDTA, 0.7% sodium Deoxycholate, 1% NP-40, 0.5 M LiCl) and one wash with $T_{50}E_{10}$ buffer (50 mM Tris-HCl pH 8.0, 10 mM EDTA) before resuspending beads in $T_{50}E_{10}S_1$ buffer (50 mM Tris-HCl pH 8.0, 10 mM EDTA, 1% SDS) and heating to 65C for 15 minutes to elute DNA. After transferring to a new tube, DNA was decrosslinked by leaving at 65C for 5 hours to overnight. DNA was treated with Proteinase K (NEB) and RNAse (Roche) before purification with phenol:chloroform:isoamyl-alcohol 1 Phase (VWR), 5Prime Phase Lock Gel Heavy tube (Quantabio), and glycogen (sigma Aldrich) to facilitate DNA pelleting. Resuspended ChIP and Input DNA was then used for qPCR or ChIP-Seq. Library preparation utilized a HyperPrep Kit (Kapa Biosystems) and NEXTFLEX DNA Barcodes (Perkin Elmer), with size selection

performed with Agencourt AMPure XP (Beckman Coulter). Libraries were sequenced on the Illumina HiSeq2000 at the MIT BioMicro Center. γH2AX ChIP-Seq was performed with three or four biological replicates.

## RNA-Seq analysis

To eliminate nucleotides that are part of the template-switching oligo as per the manufacturer's instructions (SMARTer Stranded Total RNA-Seq Kit—Pico Input Mammalian; v1 [mPFC] or v2 [HIP]; Takara), the first three nucleotides of the first sequencing read (Read 1) for kit v1 or the first three nucleotides of the second sequencing read (Read 2) for kit v2 were trimmed with Trimmomatic [110]. Trimmed reads were then aligned to the mouse genome GRCm38 (mm10) with HISAT2 [111] using default parameters. Picard MarkDuplicates (http://broadinstitute.github.io/picard/) was used to remove duplicate reads and the remaining reads sorted and indexed with SAMtools [112]. Read counts aligning to the entire gene body of each gene (introns and exons) were generated using featureCounts [113]. Analysis of differential gene expression and FPKM values were then performed with DESeq2 [114] in R, with significance determined with padj <0.05, and FPKM > 0.2 in at least one time point. To assess successful isolation of brain cell types, aggregate expression of known cell type-enriched genes was determined by taking the geometric mean of the naive FPKM values for each cell type's gene set [28], calculating the Z-score, and plotting with the R package pheatmap. To make combined signal tracks, SAMtools [112] was used to down sample replicates to an equal number of reads, before merging and generating normalized signal tracks with deepTools [115]. Genome browser signal tracks were generated with IGV [116]. Plotting was done with the ggplot2 package in R.

## ChIP-Seq analysis

ChIP-Seq reads were aligned to the mouse genome GRCm38 (mm10) with Bowtie2 [117] with default parameters. Picard MarkDuplicates (http://broadinstitute.github.io/picard/) was used to remove duplicate reads. Poorly aligned reads were then filtered out (MAPQ> 10) and the remaining reads sorted and indexed with SAMtools [112]. SAMtools [112] was used to down sample ChIP replicates to an equivalent number of reads before merging. Peaks were called with MACS2 [118] using a broad-cutoff of 0.00001. To get peaks more representative of the underlying signal, peaks were recalled with MACS2 using—broad-cutoff 0.1, and bedtools intersect [119] was used to get the overlap with the more stringently called peaks. Bedtools intersect [119] was then used to annotate peaks overlapping genes with a minimum of 50% overlap with the GRCm38.93 (mm10) GTF annotation file, with manual inspection of genome browser tracks for correction. Read counts for either gene bodies or called peaks were generated using featureCounts [113]. Analysis of differential peaks and RPKM values was then performed with DESeq2 [114] in R. MACS2 [118] was used to make read normalized (-SPMR) LogLR signal tracks and these were converted to the bigwig file format with bedGraphToBigWig [120]. Aggregate signal plots were generated with deepTools [115]. Venn diagrams of shared peaks was generated with the 'eurlerr' R package. Genome browser signal tracks were generated with IGV [116]. Plotting was done with the ggplot2 package in R.

For analysis of H3K27Ac data [57], FASTQ files were downloaded from the gene expression omnibus, accession code GSE74971. Replicates were combined before alignment and filtering, which proceeded as above. Normalized read coverage signal tacks were generated with deepTools [115].

Permutation testing was performed similar to [121]. Neuronal upregulated genes were divided into bins by expression level. Using these bins, 1000 iterations of random sampling

without replacement was conducted on all expressed genes. A weighted mean of average FPKM for the CFC 30-minute time point, accounting for the approximate neuronal (NeuN+; 60%) and non-neuronal (NeuN-; 40%) composition of whole tissue, was used as the gene expression level for each gene to allow comparison with the whole tissue γH2AX ChIP-Seq. A P-value was then calculated as the fraction of permutations that had higher mean γH2AX or RNA intensity than that for the observed upregulated genes. Plotting was done with the ggplot2 package in R.

## GO and motif analysis

To find promoter TF motif overrepresentation, the 'enricher' function of the clusterProfiler R package [21] was utilized with the MSigDB database [32] through the msigdbr R package using the category "C3" and subcategory "TFT" with a pvalueCutoff = 0.01 and all expressed genes for the pertinent condition used as background. To determine overrepresentation of biological process GOs, the 'enrichGO' function of clusterProfiler [21] was utilized with the org.Mm.eg.db Bioconductor annotation R package, with a pvalueCutoff = 0.01 and expressed genes used as background. Redundant GO categories were then removed with the clusterProfiler [21] function 'simplify', an implementation of GOSemSim [122], with similarity set at 0.7. Disease overrepresentation analysis utilized the DOSE R package [123] with a Q-value cutoff of 0.2 and expressed genes used as background. Plotting was done with the ggplot2 package in R. Clustering of related top GO terms was performed with the 'emapplot' function of the clusterProfiler [21].

## Glucocorticoid receptor external dataset analysis

Glucocorticoid receptor ChIP-Seq peaks from rat were downloaded from supplementary tables of [55, 56] and UCSC liftover [120] was used to convert the coordinates to the mouse MM10 genome. Peaks were merged with bedtools [119]. The presence of a mouse glucocorticoid receptor motif was determined by scanning the DNA sequence of each peak, obtained through the R package biomaRt [124], for the presence of the "GCR_MOUSE.H11MO.0.A" motif from the HOCOMOCO motif collection [125] with of the FIMO tool [126] of the MEME Suite [127]. GREAT [128] was used to annotate these peaks to the mouse genome, using "single nearest gene" as the annotation parameter. Plots of aggregate H3K27Ac [57] (see 'ChIP-Seq analysis') signal at glucocorticoid receptor peaks were generated with deepTools [115].

## Statistical analysis

Two-tailed unpaired student's t-test and One-way ANOVA with Tukey's multiple comparisons test were performed with GraphPad PRISM (Version 8). $P \leq 0.05$ was considered statistically significant. Bar and scatter plots show the Mean ± SEM. Outliers were detected with ROUT (Q = 2%). Other statistical tests were performed in R, including the hypergeometric distribution test using the 'phyper' function, linear regression with the 'lm' function, and Welch's ANOVA with Games-Howell post-hoc test with the 'oneway' function.

## Supporting information

**S1 Table. Sequence of primers.**
(XLSX)

**S2 Table. Genome-wide called γH2AX peaks.**
(XLSX)

**S3 Table. Nuclear RNA-Seq analysis.**
(XLSX)

**S4 Table. Glucocorticoid receptor binding sites.**
(XLSX)

**S1 Fig. γH2AX ChIP-Seq.** (A) qRT-PCR analysis of *Npas4* and *Arc* mRNA expression in the hippocampus (HIP) and medial prefrontal cortex (mPFC) 30 minutes following contextual fear conditioning (CFC30'), normalized to *Hprt* and respective naive condition. N = 4 mice per group; two-tailed unpaired student's t-test; * $P \leq 0.05$; **$P \leq 0.01$; *** $P \leq 0.001$; **** $P <0.0001$; Mean ± SEM. (B) Volcano plots of Log2FC versus Log10(FDR) of γH2AX peaks and their corresponding genes for HIP (left) and mPFC (right). Upregulated indicates FDR < 0.05 and log2FC > 0, Downregulated indicates FDR < 0.05 and log2FC < 0, ns indicates FDR > 0.05. (C) Correlation between gene length and γH2AX peak length with linear regression. Left is HIP, right is mPFC. (D) Enrichment map of the 27 enriched biological processes for the 206 γH2AX peak-containing genes shared between HIP and mPFC in Fig 1A. Over-representation analysis with gene ontology (GO) category "Biological Process." (E) Hippocampal γH2AX ChIP-qPCR analysis at the gene bodies of early response genes *Npas4*, *Nr4a1*, and housekeeping gene *B2M*. Each replicate was generated from the pooling of 3 animals' hippocampi. N = 3; IgG N = 2; two-tailed unpaired student's t-test; ns P > 0.05; * $P \leq 0.05$; ** $P \leq 0.01$; Mean ± SEM.
(TIF)

**S2 Fig. Extraction of enriched NeuN+ and NeuN- nuclei from mouse brain.** (A) Flow cytometry dot-plots representative of the gating strategy used for isolating neuronal and non-neuronal nuclei from mouse brain for RNA-Seq. Appropriately sized (FSC vs SSC), singlet nuclei (DAPI+), were gated for the presence or absence of the neuronal nuclei marker NeuN (NeuN+). (B) qRT-PCR analysis of pre-mRNA transcription from the neuronal early response gene *Npas4* and mRNA for the early response gene *Arc*. RNA purified from FACS-isolated nuclei or whole mPFC homogenate after CFC. cDNA was primed with random hexamers and the primer used for qPCR was either intronic (*Npas4*) or spanned an exon-exon junction (*Arc*). Normalized to *Hprt* and relative to naive NeuN+ for *Npas4* or normalized to respective naive condition for *Arc*. N = 3–4 mice per group; One-way ANOVA with Tukey's multiple comparisons test; absence of an asterisk indicates P > 0.05; * $P \leq 0.05$; ** $P \leq 0.01$; *** $P \leq 0.001$; **** $P <0.0001$; Mean ± SEM.
(TIF)

**S3 Fig. Nuclear RNA-seq after contextual fear conditioning.** (A) Correspondence between RNA-Seq datasets and brain cell types. Marker gene sets for brain cell types was obtained from a previously published dataset [28], and the average expression of these genes was calculated (RPKM geometric mean) for the naive conditions. Z-score determined by row. (B-E) Volcano plots of Log2FC versus log10(FDR) of RNA-Seq from HIP NeuN- (A), HIP NeuN+ (B), mPFC NeuN- (C), and mPFC NeuN+ (D). Up-regulated indicates FDR < 0.05 and log2(FC) > 0, Down-regulated indicates FDR < 0.05 and log2(FC) < 0, ns indicates FDR > 0.05. (F-G) Number of upregulated genes in neuronal (F) and non-neuronal (G) nuclei 10 to 30 minutes following CFC (RNA-Seq; FDR < 0.05). Genes shared between HIP and mPFC are in grey (CFC10') and light grey (CFC30').
(TIF)

**S4 Fig. Brain γH2AX corresponds with rapid gene induction and expression level.** (A) Per-cent overlap between genes containing a γH2AX peak and those genes upregulated

(padj < 0.05) in non-neuronal nuclei after CFC. Hypergeometric distribution test; ***
P ≤ 0.001; **** P <0.0001. (B) DSBs increase with RNA expression level; mPFC CFC30
γH2AX ChIP-Seq intensity at gene bodies versus percentile of mPFC NeuN+ CFC30' RNA
expression. Welch's ANOVA with Games-Howell post-hoc test. (C) Confirming equivalent
RNA expression levels between the observed (upregulated) genes and expected (randomly
sampled) genes for the permutation testing. Distribution of mean RNA FPKM for 1000 per-
mutations of random sampling, binned by RNA expression level. Lines are the mean RNA
FPKM. mPFC 10 minutes after CFC (left), or 30 minutes after CFC (right).
(TIF)

**S5 Fig. RNA-Seq promoter-enriched motifs in neurons.** (A-B) Top 10 enriched promoter
motifs for the genes upregulated in NeuN+ CFC10' HIP (A) and NeuN+ CFC10' mPFC (B).
Using the "Transcription Factor Targets" (TFT) gene set from the molecular signatures data-
base (MSigDB). (C) Top 8 enriched promoter motifs for the genes upregulated in neuronal
nuclei from HIP 30 minutes after CFC (Log2FC >0 & FDR < 0.05). Using the "Transcription
Factor Targets" (TFT) gene set from the molecular signatures database (MSigDB). (D) Motifs
associated with upregulated genes in HIP NeuN+ nuclei. Left, number of the indicated motifs
associated with each gene's promoter. Center, γH2AX Log2FC, right, RNA-Seq Log2 fold
change 10 or 30 minutes after CFC. Using the TFT gene sets from MSigDB for each transcrip-
tion factor motif.
(TIF)

**S6 Fig. γH2AX peaks are not enriched at late response genes.** Differentially regulated genes
from visual cortex single-cell RNA-Seq that also contain γH2AX peaks in mPFC (right), and
their regulation in mPFC nuclear RNA-Seq (left). 'Neuron' encompasses excitatory and inhibi-
tory neuron subtypes, 'Glia' includes all subtypes from oligodendrocytes, microglia, astrocytes,
and oligodendrocyte precursor cells, while 'Vasculature' denotes endothelial, pericyte, and
smooth muscle cell subtypes [42].
(TIF)

**S7 Fig. CFC induces early response genes in non-neuronal nuclei.** (A) Motifs associated with
upregulated genes in non-neuronal nuclei from HIP (top) or mPFC (bottom). Left, number of the
indicated motifs associated with each gene's promoter. Center, γH2AX Log2FC, right, RNA-Seq
Log2 fold change 10 or 30 minutes after CFC. Using the TFT gene sets from MSigDB for each
transcription factor motif. (B-C) Enrichment map of the top enriched biological processes for the
genes upregulated in HIP NeuN- (B) or mPFC NeuN- (C) nuclei 30 minutes after CFC. Over-
representation analysis with gene ontology (GO) category "Biological Process."
(TIF)

**S8 Fig. Increased HSF1 activity following CFC.** (A) HSF1 binding increases at the promoter
regions of chaperones *Hsp90ab1* and *Hspa8*. *Actb*, which is not a HSF1 target but has similarly
increased levels of γH2AX and gene expression showed lower HSF1 binding. ChIP-qPCR of
cortex 30 minutes following CFC. N = 6–7; One-way ANOVA with Tukey's multiple compari-
sons test; ns P > 0.05; * P ≤ 0.05; ** P ≤ 0.01; *** P ≤ 0.001; Mean ± SEM. (B) Increased HSF1
nuclear translocation in neurons and non-neurons after CFC in cortex. Top, representative
fluorescence intensity histograms after CFC (naïve, grey; CFC30, red). Bottom, intensity of
nuclear HSF1 after CFC was analyzed by flow cytometry; median fluorescence intensity (MFI)
normalized to respective naive condition. N = 7; two-tailed unpaired student's t-test; **
P ≤ 0.01; *** P ≤ 0.001; Mean ± SEM.
(TIF)

**S9 Fig. Related to Fig 4.** (A) qRT-PCR analysis of *Sgk1* and *Ddit4* mRNA expression in the HIP and mPFC 30 minutes following contextual fear conditioning—normalized to *Hprt* and respective naive condition. N = 4 mice per group; two-tailed unpaired student's t-test; **P $\leq$ 0.01; Mean $\pm$ SEM. (B) ChIP-qPCR analysis of γH2AX induction at select gene bodies in mouse glial primary cultures 30 minutes after treatment with dexamethasone (Dex) (100nM). N = 4 independent cultures; two-tailed unpaired student's t-test; ns P > 0.05; Mean $\pm$ SEM. (C) Average H3K27ac signal of prefrontal anterior cingulate cortex (ACC) at glucocorticoid receptor binding sites (rat cortical ChIP-Seq) [55,56] containing the GC motif in mouse (n = 5591 peaks). H3K27ac ChIP-Seq of naive mouse ACC from NeuN+ or NeuN- isolated nuclei [57]. (D) Average H3K27ac signal at intergenic glucocorticoid receptor binding sites (rat cortical ChIP-Seq) [55,56] containing the GC motif in mouse (n = 1860 peaks). H3K27ac ChIP-Seq of mouse hippocampal CA1 region from NeuN+ or NeuN- isolated nuclei 1 hour after exposure to context, or CFC (shock) [57]. (E) Mean expression of the glucocorticoid receptor (*Nr3c1*) from naive nuclear RNA-Seq datasets. (F) Intensity of nuclear glucocorticoid receptor (left) or nuclear NeuN (right) in NeuN+ or NeuN- nuclei of the mPFC after corticosterone treatment. Normalized median fluorescence intensity (MFI). N = 5 mice per group; two-tailed unpaired student's t-test; ** P $\leq$ 0.01; ns P > 0.05; Mean $\pm$ SEM. (TIF)

**S10 Fig. Purification and confirmation of glia cell types.** (A) Flow cytometry dot-plots representative of the gating strategy used for isolating neuronal and glial nuclei from hippocampus for RNA-Seq. Appropriately sized (FSC vs SSC plot), singlet nuclei (DAPI+; DNA stain), are gated for the presence or absence of the neuronal nuclei marker NeuN (NeuN+); NeuN- nuclei are then separated by the astrocyte marker GFAP (GFAP+), GFAP- nuclei are then gated for the microglia marker P.1 (PU.1+), with the oligodendrocyte-enriched fraction, NeuN-/GFAP-/PU.1- (3X-), also collected. (B) RT-qPCR analysis of FACS-isolated nuclei for cell-type-specific markers: neuronal (*Npas4*), astrocyte (*Gfap*), microglial (*C1qa*), oligodendrocyte (*Mbp*). Normalized to *Hprt*; N = 8, 4 mice saline treated, 4 mice corticosterone treated. (C) Glucocorticoid receptor agonist corticosterone (Cort) induces *Sgk1*, *Ddit4*, and *Glul* in glia, and not the housekeeping gene *Actb*. RT-qPCR on FACS-purified hippocampal nuclear RNA from neurons (NeuN+), astrocytes (GFAP+), microglia (PU.1+), and oligodendrocytes-enriched (NeuN-GFAP-PU.1-; 3X-) 30 minutes after saline or corticosterone:HBC complex (2mg/Kg) administration. N = 4 mice per group; two-tailed unpaired student's t-test; * P $\leq$ 0.05; *** P $\leq$ 0.001; **** P <0.0001; Mean $\pm$ SEM.; * P $\leq$ 0.05; *** P $\leq$ 0.001; **** P <0.0001. (TIF)

**S11 Fig. Extensive corticosterone-mediated gene induction in the hippocampus.** (A) Correspondence between hippocampal cell-type-specific RNA-seq datasets and brain cell types. Marker gene sets for brain cell types was obtained from a previously published dataset, and the average expression of these genes was calculated (RPKM geometric mean) for the saline condition [28]. Z-score determined by row. (B-E) Volcano plots of Log2FC versus Log10(FDR) of RNA-Seq from corticosterone treated hippocampal cell types. Upregulated indicates FDR < 0.05 and log2(FC) > 0, Downregulated indicates FDR < 0.05 and log2(FC) < 0, ns indicates FDR > 0.05. (TIF)

**S12 Fig. Corticosterone induced biological process GO terms.** (A-C) Enrichment map of the top 30 enriched biological processes for the genes upregulated in HIP after corticosterone treatment in astrocyte (A), oligodendrocyte-enriched (B), and microglia (C) nuclei. No enrichment in neurons. Over-representation analysis with gene ontology (GO) category "Biological

Process."
(TIF)

**S13 Fig. Corticosterone repressed biological process GO terms.** (A-C) Enrichment map of the top enriched biological processes for the genes downregulated in HIP after corticosterone treatment in astrocytic (A), oligodendrocyte-enriched (B), and microglial (C) nuclei. Over-representation analysis with gene ontology (GO) category "Biological Process." (D) Top five enriched biological processes of the downregulated genes for each cell type in purified hippo-campal nuclei from neurons, astrocytes, microglia, and oligodendrocyte-enriched after corti-costerone treatment (padj < 0.05). Over-representation analysis with gene ontology (GO) category "Biological Process."
(TIF)

**S14 Fig. γH2AX peak at site of convergent transcription within gene *Polr3e*.** Genome browser tracks for the gene *Polr3e* displaying a small (951bp) intronic γH2AX peak in mPFC overlapping a mammalian interspersed repeat (MIR). MIR antisense transcription is shown on the negative strand (-).
(TIF)

**S15 Fig. Enriched disease associations after corticosterone treatment.** Enriched disease associations for the genes upregulated in HIP after corticosterone treatment in neuronal, astrocytic, microglial, and oligodendrocyte-enriched nuclei, filtered for disease associations related to the nervous system. An absent group indicates no enrichment at threshold q < 0.05. Over-representation analysis with DOSE Disease ontology.
(TIF)

## Acknowledgments

We thank, Jay Penney, Hugh Cam, Matheus Victor, Omer Durak, Ping-Chieh Pao, Vishnu Dileep, and Ram Madabhushi for thoughtful comments and feedback on the manuscript; Jem-mie Cheng for nuclear FACS optimization; Ram Madabhushi and Jemmie Cheng for ChIP guidance; Ping-Chieh Pao, Audrey Lee, and Chinnakkaruppan Adaikkan for experimental assistance; Ying Zhou for laboratory management; Erica McNamara for maintaining the mouse colony; the members of the Tsai laboratory for feedback and advice on this project.

## Author Contributions

**Conceptualization:** Ryan T. Stott, Li-Huei Tsai.

**Data curation:** Ryan T. Stott.

**Formal analysis:** Ryan T. Stott.

**Funding acquisition:** Li-Huei Tsai.

**Investigation:** Ryan T. Stott, Oleg Kritsky.

**Methodology:** Ryan T. Stott.

**Project administration:** Li-Huei Tsai.

**Supervision:** Li-Huei Tsai.

**Validation:** Ryan T. Stott.

**Visualization:** Ryan T. Stott.

**Writing – original draft:** Ryan T. Stott.

**Writing – review & editing:** Ryan T. Stott, Oleg Kritsky, Li-Huei Tsai.

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
