## [Decision Letter · Decision Letter 0]

13 Apr 2021

PONE-D-21-08714

Profiling DNA break sites and transcriptional changes in response to contextual fear learning

PLOS ONE

Dear Dr. Tsai,

Thank you for submitting your manuscript to PLOS ONE. Two experts have reviewed your manuscript, and I pasted their comments below. Both reviewers expressed substantial enthusiasm toward you manuscript, but they also raised a few points that need to be addressed. Therefore, we invite you to submit a revised version of the manuscript that addresses the points raised during the review process.

We look forward to receiving your revised manuscript.

Kind regards,

Bing Yao

Academic Editor

PLOS ONE

Journal Requirements:

PLOS ONE has specific criteria regarding the reporting of animal research (https://journals.plos.org/plosone/s/submission-guidelines#loc-animal-research). Specifically, these guidelines require that details regarding care, monitoring, and method of sacrifice are clearly stated. To that effect, please include in your Methods section the total number of animals used and the method of anesthesia used (if applicable).

As part of your revision, please complete and submit a copy of the ARRIVE Guidelines checklist, a document that aims to improve experimental reporting and reproducibility of animal studies for purposes of post-publication data analysis and reproducibility: https://arriveguidelines.org/sites/arrive/files/Author%20Checklist%20-%20Full.pdf. Please include your completed checklist as a Supporting Information file. Note that if your paper is accepted for publication, this checklist will be published as part of your article.

Reviewers' comments:

Reviewer's Responses to Questions

**Comments to the Author**

1. Is the manuscript technically sound, and do the data support the conclusions?

Reviewer #1: Yes

Reviewer #2: Yes

2. Has the statistical analysis been performed appropriately and rigorously? 

Reviewer #1: Yes

Reviewer #2: Yes

3. Have the authors made all data underlying the findings in their manuscript fully available?

Reviewer #1: Yes

Reviewer #2: Yes

4. Is the manuscript presented in an intelligible fashion and written in standard English?

Reviewer #1: Yes

Reviewer #2: Yes

5. Review Comments to the Author

Reviewer #1: Unlike cycling cells, where DNA replication and cell cycle progression respond to DSBs and activate the repair mechanism, neuron cells do not proliferate. Hence, it is intriguing how neuron cells handle DSB damage and moreover how DSB formation interplays with the neuronal activities. In this manuscript by Stott et.al, the authors mapped the genome-wide DSB formation in the mouse prefrontal cortex and hippocampus under contextual fear conditioning, which provided an interesting connection between learning behaviors and DNA damage in the brain. I found this research very interesting and technically sound, therefore, recommend its acceptance for publication at PLOS ONE.

Reviewer #2: In this study, the authors investigate fear conditioning induced DNA double strand breaks (DSBs) in adult male mouse hippocampus and medial prefrontal cortex through ChIPseq of γH2AX. They recognize widely distributed DSBs after fear conditioning with many demonstrated brain region specific inductions. These DSBs are not only enriched in a few functionally relevant gene categories (e.g. synaptic transmission), but also correlated with transcription changes in neurons and/or glia cells which are profiled through cell-type specific nuclear RNAseq. Furthermore, the upregulated transcripts appear to have higher enrichment of γH2AX and are mediated through activity-dependent gene related transcription factors, which further indicates the functional role of DSBs in neural plasticity. Notably, several fear conditioning induced genes that carry DSBs are modulated by heat shock factor, which suggests an involvement of proteostasis in fear conditioning induced DSBs. Interestingly, many of the γH2AX-containing genes are responsive to glucocorticoid only in non-neuronal nuclei. Following corticosterone administration, the authors find that the differential genes are predominantly in glial cells, with some of them showing high levels of γH2AX. The study includes a large amount of work and the findings are novel. I only have a few minor comments.

1. The study has used γH2AX as a DSB proxy and performed its ChIPseq to identify DSBs genome-wide. It will be helpful to discuss if any limitations of using γH2AX to recognize DSBs. Will DNA single strand break also be detected through this approach?

2. Line 82, (B): “six” should be “five”.

3. Line 85, (C): Double check if it is top “5” biological processes… or more.

4. Ling 334: “GC” should be “GR”.

5. Figure 3A, y-axis: It is unclear why some HSF motifs are shown with a sequence, some are not.

6. PLOS authors have the option to publish the peer review history of their article (what does this mean?). If published, this will include your full peer review and any attached files.

Reviewer #1: No

Reviewer #2: No

---

## [Author Response · Author response to Decision Letter 0]

25 May 2021

Response to Reviewers

We would like to thank the reviewers for their time and input on this manuscript. We appreciate their comments and suggestions. Please see our responses in red. 

Reviewer #1: Unlike cycling cells, where DNA replication and cell cycle progression respond to DSBs and activate the repair mechanism, neuron cells do not proliferate. Hence, it is intriguing how neuron cells handle DSB damage and moreover how DSB formation interplays with the neuronal activities. In this manuscript by Stott et.al, the authors mapped the genome-wide DSB formation in the mouse prefrontal cortex and hippocampus under contextual fear conditioning, which provided an interesting connection between learning behaviors and DNA damage in the brain. I found this research very interesting and technically sound, therefore, recommend its acceptance for publication at PLOS ONE.

We thank the reviewer’s comment.

Reviewer #2: In this study, the authors investigate fear conditioning induced DNA double strand breaks (DSBs) in adult male mouse hippocampus and medial prefrontal cortex through ChIPseq of γH2AX. They recognize widely distributed DSBs after fear conditioning with many demonstrated brain region specific inductions. These DSBs are not only enriched in a few functionally relevant gene categories (e.g. synaptic transmission), but also correlated with transcription changes in neurons and/or glia cells which are profiled through cell-type specific nuclear RNAseq. Furthermore, the upregulated transcripts appear to have higher enrichment of γH2AX and are mediated through activity-dependent gene related transcription factors, which further indicates the functional role of DSBs in neural plasticity. Notably, several fear conditioning induced genes that carry DSBs are modulated by heat shock factor, which suggests an involvement of proteostasis in fear conditioning induced DSBs. Interestingly, many of the γH2AX-containing genes are responsive to glucocorticoid only in non-neuronal nuclei. Following corticosterone administration, the authors find that the differential genes are predominantly in glial cells, with some of them showing high levels of γH2AX. The study includes a large amount of work and the findings are novel. I only have a few minor comments.

1. The study has used γH2AX as a DSB proxy and performed its ChIPseq to identify DSBs genome-wide. It will be helpful to discuss if any limitations of using γH2AX to recognize DSBs. Will DNA single strand break also be detected through this approach?

We agree this is an important point that should be addressed. We have added a new paragraph in the discussion to address the limitations of γH2AX ChIP-seq (please see lines 442-454).

2. Line 82, (B): “six” should be “five”.

We count six for this figure, please see Fig 1B below which is being referenced on line 82:

3. Line 85, (C): Double check if it is top “5” biological processes… or more.

We double-checked and found that we had indeed selected the top five biological processes from each of the four time points.

4. Ling 334: “GC” should be “GR”.

We have corrected this mistake and a few other instances, including in Fig 4.

5. Figure 3A, y-axis: It is unclear why some HSF motifs are shown with a sequence, some are not.

We utilized the given name for each gene set without editing, and some of these names include motifs for reasons unclear. We agree it is a bit confusing, but we did not want to edit them because it could obscure their identity.

---

## [Editor Report · Decision Letter 1]

28 May 2021

Profiling DNA break sites and transcriptional changes in response to contextual fear learning

PONE-D-21-08714R1

Dear Dr. Tsai,

We’re pleased to inform you that your manuscript has been judged scientifically suitable for publication and will be formally accepted for publication once it meets all outstanding technical requirements.

Kind regards,

Bing Yao

Academic Editor

PLOS ONE
---

## [Editor Report · Acceptance letter]

23 Jun 2021

PONE-D-21-08714R1 

Profiling DNA break sites and transcriptional changes in response to contextual fear learning 

Dear Dr. Tsai:

I'm pleased to inform you that your manuscript has been deemed suitable for publication in PLOS ONE. Congratulations! Your manuscript is now with our production department. 

Kind regards, 

on behalf of

Dr. Bing Yao 

Academic Editor

PLOS ONE